# Tumor Lung Visualization and Localization through Virtual Reality and Thermal Feedback Interface

**DOI:** 10.3390/diagnostics13030567

**Published:** 2023-02-03

**Authors:** Samir Benbelkacem, Nadia Zenati-Henda, Nabil Zerrouki, Adel Oulefki, Sos Agaian, Mostefa Masmoudi, Ahmed Bentaleb, Alex Liew

**Affiliations:** 1Division Robotique et Productique, Centre de Développement des Technologies Avancées, Baba Hassen 16081, Algeria; 2Computer Science Department, City University of New York, New York, NY 10314, USA; 3Département Image et Traitement de l’Information, Institue Mines-Télécom (IMT) Atlantique, 29238 Brest, France

**Keywords:** virtual reality (VR), thermal model, thermal sensation, tumor localization

## Abstract

The World Health Organization estimates that there were around 10 million deaths due to cancer in 2020, and lung cancer was the most common type of cancer, with over 2.2 million new cases and 1.8 million deaths. While there have been advances in the diagnosis and prediction of lung cancer, there is still a need for new, intelligent methods or diagnostic tools to help medical professionals detect the disease. Since it is currently unable to detect at an early stage, speedy detection and identification are crucial because they can increase a patient’s chances of survival. This article focuses on developing a new tool for diagnosing lung tumors and providing thermal touch feedback using virtual reality visualization and thermal technology. This tool is intended to help identify and locate tumors and measure the size and temperature of the tumor surface. The tool uses data from CT scans to create a virtual reality visualization of the lung tissue and includes a thermal display incorporated into a haptic device. The tool is also tested by touching virtual tumors in a virtual reality application. On the other hand, thermal feedback could be used as a sensory substitute or adjunct for visual or tactile feedback. The experimental results are evaluated with the performance comparison of different algorithms and demonstrate that the proposed thermal model is effective. The results also show that the tool can estimate the characteristics of tumors accurately and that it has the potential to be used in a virtual reality application to “touch” virtual tumors. In other words, the results support the use of the tool for diagnosing lung tumors and providing thermal touch feedback using virtual reality visualization, force, and thermal technology.

## 1. Introduction

Lung cancer is a type of cancer that affects the lungs and is characterized by the uncontrolled growth of abnormal cells. These cells can disrupt the normal functioning of the lungs, which are responsible for providing oxygen to the body through the blood [1,2,3]. Lung cancer has a high mortality rate and is often not easily curable at advanced stages, making early detection crucial. Even if there are many advances in treatment strategies, lung cancer at an advanced or late stage is not often easily curable [4], so detecting lung cancer nodule regions in the earlier stage is crucial.

Innovative diagnostic tests and screening methods, such as CT scans and MRI, are essential tools for detecting lung cancer and providing healthcare professionals with information they can use to make accurate decisions [5]. However, these methods can be time-consuming and costly and may not always provide the necessary information for accurate decision making. One approach for detecting cancer tumors in CT scan imagery involves using algorithms to cluster and analyze the imagery, identify disconnected areas, and locate tumors using active contour algorithms [6]. Studies have shown that involving expert doctors and radiologists is necessary for accurately diagnosing and detecting lung cancer nodules. However, the symptoms of lung cancer often do not appear until the disease has progressed to a severe stage, at which point it may be difficult to cure. This is partly due to the time it takes to diagnose and treat lung cancer and the lack of recognition of symptoms at an early stage. Improving the speed of diagnosis and treatment is vital for increasing the chances of curing lung cancer. Screening for lung cancer is vital in improving detection and prognosis tools and patient health. The CT scan images incorporate a tremendous amount of information about nodules, and an increasing number of images make their accurate assessment challenging for radiologists [7]. Additionally, using images to detect cancer can lead to false negatives and positives, which can cause tension, second opinion, and additional costs for patients and doctors. Using novel computational technology in lung cancer detection can improve accuracy and performance compared to human performance. Recently, various methods have evolved based on handcraft and learned approaches to assist radiologists. For more about CT images-based lung cancer detection, classification, identification, and the comprehensive analysis of different methods, one can refer to the following review papers [7,8]. Additionally, the CT scan image-based diagnosis methods are not practical for regular mass screening in short intervals.

One solution to this issue could be thermography. Thermography is a diagnostic technique that uses a special camera to produce images showing the temperature of different body parts. These images can help healthcare professionals detect abnormalities indicative of certain medical conditions. The advantages of thermography are that (i) it is non-invasive, meaning it does not involve using any instrument or device that touches the body, (ii) it is a relatively low-cost and safe option for mass screening, as it can be done quickly and without the need for specialized equipment, and (iii) it can detect temperature changes in the body that certain medical conditions might cause. For example, tumors often cause changes in the temperature of the surrounding tissue, which can be detected using thermography. This makes it a valuable tool for the early detection of certain conditions, improving a person’s chances of receiving timely and effective treatment. Additionally, thermography may offer a new approach to detecting abnormalities in tissue and providing temperature profile information to assist practitioners in diagnosing and treating lung cancer.

Another solution could be using virtual reality (VR) and augmented reality (AR) technologies. VR uses a computer to simulate a three-dimensional environment that can be interacted with in a seemingly real or physical way. AR is a technology that superimposes a computer-generated image on a user’s view of the real world, providing a composite view. Both have the potential to be used in healthcare, such as Alzheimer’s disease [9], diagnostics and therapy of neurological diseases [10], dermatology [11] and surgical planing [12]. Additionally, AR can provide real-time information to healthcare professionals during procedures, allowing them to access patient data and make informed decisions. It can also educate patients about their condition and treatment options. VR can train medical professionals in simulated environments, allowing them to practice procedures and gain experience without the risk of harm to actual patients. VR can also provide therapy for patients with phobias or post-traumatic stress disorder (PTSD). So, combined thermography, VR, and AR technologies may improve healthcare delivery quality and efficiency and make it more accessible to people in remote or underserved areas.

Several works have utilized VR technology to address the visualization of lung cancer through VR. Yoon [13] annotated lung tumor computed tomography images and illustrated them in a three-dimensional (3D) printed airway model. The lung tumor was localized by computed tomography (CT) driven virtual reality (VR) endoscopy. Authors in [14] introduced VR in a smart operating room of Seoul National University Bundang Hospital in Korea for training in lung cancer surgery. In [15], virtual reality was combined with artificial intelligence techniques for pulmonary segmentectomies planning to well represent and visualize lung cancer. In the same sense, works presented in [16,17,18] studied the added clinical value of three-dimensional (3D) based VR for preoperative planning and assisted thoracic surgery. VR was also combined with cloud computing to accelerate the data annotation of a patient with lung cancer.

Despite significant advances in segmentation and diagnosis methods developed for lung tumor localization, there is still room for further research and exploration. One of the current challenges is introducing a three-dimensional (3D) framework for developing advanced lung tumors 3D visualization and localization using VR. The work carried out allows the visualization of the tumor. However, the tumor cannot be well localized upon visualization. A possible solution could be providing thermal feedback to the user to feel the tumor’s location better. This paper offers a tumor lung visualization and localization through virtual reality and thermal feedback interface to address the above-mentioned challenges. The primary goal of this system is to provide a more effective and efficient way to detect and analyze lung tumors, improving the accuracy and speed of diagnosis and treatment. The main contributions are the following:Presents theoretical and experimental studies of a model describing the heat transfer occurring when a finger touches a tumor using the lung cancer CT-scan imagery transformed into 3D visualization data via VR technology.Advanced virtual-reality-based 3D visualization and interaction system for lung tumor recognition, quantification, and analysis.Develop an interface for thermal rendering and thermal identification of the virtual touched object, as well as utilizing a thermal display incorporated into a haptic device.

The paper also presents experimental results that validate the use of this VR system for analyzing tumors in virtual environments. The remaining part of the paper is organized as follows: Section 1 reviews the related work on lung tumor diagnostic systems. Section 2 provides the proposed method of thermal exchange approaches, including the 3D visualization and interaction. Section 3 presents the experimental results, and Section 4 describes the paper’s discussion and conclusions.

## 2. Materials and Methods

A detailed discussion of the proposed method is presented in this section. We developed lung cancer identification and localization platform, composed of three major modules: the 3D reconstruction module (3D-RM), VR interaction module (VRIM), and 3D localization module (3DL) as shown in Figure 1.

Firstly, the 3D-RM module inputs CT-scan imagery to delineate and 3D reconstruct the tumor inside the lung. Through surface models of segmented 2D images, data volumes were generated by applying iso-surface extraction marching cubes and data volume-rendering algorithms [19,20]. Secondly, the role of the VRIM module is to visualize in VR the classified part tumor/no tumor. We integrated virtual reality with CT-scan imagery to generate a three-dimensional and realistic display of lung tumors. We used Unity3D or Unreal Engine as support to generate 3D models. We implemented a 3D interaction algorithm through a data-glove in order to manipulate 3D models VR environment. Lastly, the 3DL module provides a thermal feedback of the volumetric model displayed of the lung tumors using a thermal interface composed of Peltier devices. This module allow users to obtain thermal sensation of the tumor. With this technique, the tumor’s size could be well perceived.

In our case, we used algorithms developed in [2,21,22] to implement segmentation and 3D reconstruction parts of lung cancer. The results obtained are not the subject of this paper. Thus, the 3D-RM module will not be presented. We, essentially, focused on the VR part which includes VRIM and 3DL modules.

### 2.1. Virtual Reality Visualization and Interaction for Lung Tumor Region Segmentation

In this section, the proposed lung tumor region segmentation and VR visualization and interaction used are presented. Virtual reality was considered with CT-scan imaging to provide a comprehensive display of lungs with tumors detailed lesions. The results obtained from the segmentation [2,21,22,23] of CT-scan images are used as processed data to generate a 3D sub-region tumor visualization and interaction system. Although our segmentation method provides a reasonable segmentation, the method proposed is less time-consuming and does not require a high-performance graphical card. This makes it possible to perform segmentation and 3D visualization faster. Our approach could be suitable for practical situations. Radiologists, in hospitals and clinics, should have the results as quickly as possible in order to deal with a high number of patients, provide adapted treatment, facilitate management of disease and reduce complex situation.

In order to improve the medical diagnostic strategy, we provide the radiologists an interactive VR tool that interactively visualizes the infected lungs in 3D.

Figure 2 shows the general diagram developed in the Unity game engine. It illustrates the main components of the VR application that developed using Unity 3D/Unreal Engine. In this case, we integrated several packages: the virtual environment package, which integrated optional 3D objects of the scene, such as walls, tables, lamps, paintings, etc.; three packages for 3D lungs design (3D lung models, 3D bronchus models, and 3D tumor models); interaction package for human–computer interaction management; and data manager package for data exchange between packages and 3D scene updates.

The hardware part is composed of the Oculus Quest Head Mounted Display (HMD) (version updated of Oculus Rift [24]) connected to the computer via the DisplayPort. It integrates sensors to recognize the user’s head movements in the space. Oculus Touch, mouse, and keyboard, manage the 3D interaction between the user and the VR application. The user manipulates the 3D lungs tumor and goes inside the lungs to view the 3D tumor in more details.

### 2.2. Modeling Heat Exchange for Simulation

This section introduces the thermal exchange simulation and how it was developed for the lung tumor application. However, other approaches, including Pennes’ bio heat transfer, have been used to solve the issue of thermal exchange simulation. When we want this thermal simulation to be similar to or as close to the real-world environment as possible, however, simulating heat exchange remains laborious. In other words, the temperature of the cells impacted by the tumor is higher than that of the surrounding cells due to increased blood flow into the tumor, and the thermal transfer model has a significant influence on the outcome [25].

In the present work, the modeling heat transfer of lung tumor is based on Pennes’ bio technique. The capacity of this Penne’s bioheat transfer model to estimate temperature change within human tissues was the primary factor in its selection. Additionally, the latter takes into account the impacts of temperature changes associated to the diffusion and perfusion of blood into tissues, in addition to the heat exchange that occurs between tissues and blood flow [26]. The following equation can be used to express Penne’s bioheat transport model of a tumor:(1)$$\rho_{org}\varsigma_{org}\frac{\gamma \tau_{org}}{\gamma t}=\lambda_{org}\frac{\gamma \tau_{org}}{\gamma x^{2}}+m_{b}c_{b}\left(\varsigma_{\alpha r}-\varsigma_{org}\right)+Q_{org}$$
where $\tau_{ar}$ and $\tau_{org}$ represent arterial, and organ (tumor) temperatures, and $\rho_{org}$, $\varsigma_{org}$, and $\gamma_{org}$ represent density, specific heat, and thermal conductivity of the organ (tumor) tissue, respectively. $m_{b}$ and $c_{b}$ indicate the perfusion speed and the specific heat of blood, and $Q_{org}$ denotes the metabolic heat generation term. Otherwise, the thermal therapy is considered a heat source where the heat flux ϕ is supposed to be constant. Mathematical aspects of boundary conditions are given as follows:


(2)
$$\left\{\begin{array}{c} -\lambda_{org}\frac{\delta \tau^{org}\left(0,t\right)}{\delta x}=\phi \\ \tau_{org}(x,0)={}_{i}^{org}\\ \tau_{org}(\omega ,0)={}_{i}^{org} \end{array}\right.$$


$\tau_{org}^{i}$ represents the initial organ (tumor) temperature.
(3)$$\frac{\delta \tau_{org}}{\delta_{t}}=\alpha_{org}\frac{\delta^{2}\tau_{org}}{\delta^{2}}-a\tau_{org}+a\tau_{ar}+bQ_{org}$$
where
(4)$$a=\frac{m_{b}c_{b}}{\rho_{org}C_{org}}\&b=\frac{1}{\rho_{org}C_{org}}$$

Let us denote
(5)$${\tilde{T}}_{org}=T_{org}-T_{org}^{i}$$
(6)$$\frac{\delta {\tilde{T}}_{org}}{\delta_{t}}=\alpha_{org}\frac{\delta^{2}{\tilde{T}}_{org}}{\delta^{2}}-a{\tilde{T}}_{org}+a\left(T_{ar}-T_{org}^{i}\right)+bQ_{org}$$

By applying Laplace transform, we obtain
(7)$$s\theta_{org}\left(x,s\right)=\alpha_{org}\frac{\delta^{2}\theta_{org}\left(x,s\right)}{\delta x^{2}}-a\theta_{org}\left(x,s\right)+\frac{a\left(T_{ar}-T_{org}^{i}\right)+bQ_{org}}{s}$$
where $\alpha_{org},s,\&\theta_{org}$ represent the thermal diffusivity in the organ, Laplace variable, and Laplace transform of ${\tilde{T}}_{org}$ respectively.

By combination of homogeneous $\theta_{org}^{h}$ particular $\theta_{org}^{p}$ solutions, one can denote the general solution as follows:(8)$$\theta_{org}\left(x,s\right)=\frac{c\beta_{org}+\phi {\sqrt{s+a}}^{-(\sqrt{\frac{s+a}{\alpha_{org}}})}}{\beta_{org}s\left(s+a\right)}$$

When applying the Gaver–Stehfest algorithm [27] (with N=10), the resulting function can expressed as
(9)$$T_{org}\left(x,t\right)=\frac{ln(2)}{t}\sum_{J=1}^{N}V_{j}\theta_{org}\left(x,\frac{jln(2)}{t}\right)+T_{org}^{i}$$
where V=[0.0833333333,−32.08333333,1279.000076,−15623.66689,84244.16946,−236957.5129,375911.6923,−340071.6923,164062.5128,−32812.50256].

Table 1 below shows different proprieties of tumor tissues.

It is well established that the value of temperature at the tumor tissue is around 40 °C [29], where the variation of initial temperature values, namely (T1 = 39°, T2 = 40°, T3 = 41°). From Figure 3, one can observe the proportionality of the variations of temperature compared to the initial values, where the temperature of lung tumor converges toward a value slightly higher than the initial value. The perceived temperature difference is relatively homogeneous and varies approximately between 1 °C and 3 °C from the initial temperature. In a second part, to perceive and estimate the thermal sensation during the interaction with the tumor through the glove, we were interested in the study regarding the variation of the thermal flux. It is well known that the heat flux exchange is considered a significant indicator of the heat perceived by the patient [26]. For this, we applied several sizes or thicknesses of tumors to see the variation of the heat flux (as illustrated in Figure 4). Three different values of size have been considered, namely 2.5 mm, 3 mm, and 3.3 mm. As first reflection (from Figure 4), one can note that the heat flux is inversely proportional to the tumor thicknesses. One can also notice the presence of a first short phase (does not exceed 10 s) characterized by a considerable increase, then a second phase of convergence.

The results obtained on the tumor thermal model were used to simulate its thermal behavior in virtual reality.

### 2.3. Thermal Tactile Display and Virtual Reality for Lung Tumor Visualization and Localization

For lung tumor visualization, the Oculus Quest headset was used to provide virtual reality for the wearer. Virtual reality (VR) headsets comprise a stereoscopic head-mounted display (providing separate images for each eye), stereo sound, and head motion-tracking sensors (which may include gyroscopes, accelerometer, magnetometers, structured light systems, etc.). Some VR headsets also have eye-tracking sensors and gaming controllers.

For lung tumor localization, we integrated to our virtual reality (VR) headset an interactive glove to design a thermal display associated to a haptic interface. Our proposed device is shown in Figure 5. The front view (see Figure 5a) emphasizes the glove’s structure, while the side view (see Figure 5b) emphasizes the 3D organ manipulation.

Researchers have proposed different prototypes for smart wearables in virtual reality. The developed bracelets were managed by heavyweight, bulkiness and a wired external control system. Additionally, few dimensions of sensations were provided to users. In our case, we proposed the HapLeap optimized wearable grasping device (see Figure 6) to stimulate the user’s hand with three types of sensations: kinesthetic, cutaneous and thermal feedbacks. HapLeap is a real-time hand thermo-haptic glove for stimulating the physical materials behavior in a virtual environment. Hapleap is equipped with vibrotactile tactors that generate vibrations through user’s fingers when a 3D tumor inside a 3D infected lung of a patient is touched and/or grasped. For the second situation, the fingers motion will be blocked to simulate the force effect on the hand. For the thermal effect, Peltier modules are used to simulate the temperature of the object being interacted with. Thus, the thermal display is integrated to the glove’s fingertip, allowing the user to feel thermal sensation (cold and hot) when touching and/or grasping a 3D tumor corresponding to a patient.

Our virtual environment includes 3D infected lungs, including tumors. The thermal behavior of tumors is generated using the thermal model developed in Equations (8) and (9). The simulation algorithm is quite simple: the fingertip’s position is given by the glove and is sent to the virtual environment. A collision detection checks for the proximity distance between the virtual finger model and the 3D tumor. Once a contact is determined, the touched 3D tumor is identified, and the predicted temperature of the sensor is computed by our model. This temperature is then sent as the desired temperature to be displayed to the user’s finger through Peltier modules.

To control the tactors, pulse width modulated (PWM) waveform was used. When a user interacts with a 3D tumor, the surface type value is transmitted to HapLeap and processed with the control unit. The vibration intensity is then controlled by driving the tactors with a series of ON/OFF pulses and varying the duty cycle (ratio of the time that the output is ON compared to when the output is OFF), while keeping the frequency constant. The longer the pulse ON, the faster the tactors will rotate. This leads to rough vibrations. Likewise, the shorter the pulse ON, the slower the motor will rotate, which leads to softer vibrations. We associate the haptic feedback to the index and thumb.

Based on the similar concept of PWM signal, the Peltier modules are controlled with another generated signal from the micro-controller. Additionally, the Peltier control signal is affected by a closed loop control system from the micro-controller to maintain a reasonable temperature value that will not exceed some maximum value for safety. This control system is realized using a temperature sensor placed between the Peltier and the finger skin. The current temperature is transmitted to the control unit as well as the “user-3D tumor” interaction data received via Bluetooth. The generated signal is connected to the L293D integrated circuit. The latter is a typical motor driver that allows 2 DC motors to drive in two directions. The first driver channel is used for the Peltier with two directions, as the first direction is used to heat up the finger’s area and the second direction is used to cool down finger’s area for faster and reliable sensation of the 3D tumor. Finally, we set the thermal feedback for the index and thumb.

## 3. VR Application Results

We developed a virtual reality (VR) platform that allows the visualization, and manipulation of 3D data from medical imaging sources. In practice, a stack of DICOM imagery was transformed to [obj] and/or [slt] files. In our setup, we used the Blender software framework [30] to input [obj] files and directly generate the FBX format for lung cancers visualization and localization. With Unity 3D [31] or Unreal Engine [32] engines, we generate VR-infected lungs in which tumors are well segmented.

Figure 7 and Figure 8 show a radiologist visualizing lung tumor with a VR interface through multi-view access, using an Oculus Quest head mounted display (HMD). The experimenter can even navigate into the 3D lungs and see more details on the 3D lesion texture. He also can touch the tumor using the HapLeap controller. Haptic and thermal interfaces are switched on as soon as the tumor is localized.

When the surface of the 3D tumor is touched, from any point of contact (Figure 8) by the virtual hand, the subject feels vibration (haptic sensation) and a temperature close to 40 °C, which corresponds to the temperature of the tumor. When the virtual hand is far from the 3D tumor (Figure 7), no temperature is felt. Consequently, the experimenter can localize and quantify the tumor zone inside the lung.

## 4. Participants

Eight (08) members of a medical team composed of radiologists, doctors and students agreed to test our lung cancer VR visualization and localization system and report their user experience in a subjective survey. In order to analyze doctors’ interactions with 3D realistic diseased lungs in immersive virtual worlds, a survey was prepared with VR experts. For realistic immersion, we tracked the user’s head and hand movements using the Oculus Quest Head Mounted Display (HMD). Tablets and smartphones could also be used for this experiment.

## 5. Procedure

The medical personnel wore the HMD and watched patients’ 3D infected lungs for eight minutes, though this time might be increased if necessary. The duration of immersion was recorded. The VR-naive examiner was not given any training or other explanation after that. Participants were not disturbed or swayed when filling out a subjective questionnaire following the event. In the testing process, participants were required to identify the lesion characteristics by contrasting VR lung tumor models with scan medical images in terms of volume, location and its spread. The subjects were asked to complete a short survey about their experiences and a questionnaire graded on a seven-point Likert scale when the trials were finished. On a seven-point scale, participants were asked to rate their agreement or disagreement with 07 statements about the VR diagnostic platform. The dataset was also examined to see if medical staff volunteers who had never experienced VR had a high level of comfort. In this instance, it is conceivable to research how effective the VR application is for radiologists under both normal working circumstances and extraordinary ones. Participants were then asked to list the qualities they thought were the best. The experimental parameters that need to be evaluated are parameters from 04 to 07 (inspired from [33]). In the meanwhile, the authors proposed the following parameters from 01 to 03:(Q${}_{1}$) It is easy to imagine being completely submerged in a 3D infected lung.(Q${}_{2}$) Tumor is well localized.(Q${}_{3}$) The application is easy to use.(Q${}_{4}$) Increase comprehension of and knowledge of disease.(Q${}_{5}$) Provide a realistic view of clinical case.(Q${}_{6}$) Could help reduce error.(Q${}_{7}$) An enjoyable experience.

## 6. Discussion

According the experimenters survey, seven (07) participants felt completely submerged when they manipulated the 3D lung. Five (05) users expressed surprise and agreed. On the other hand, the majority of subjects found the 3D lung tumors realistic and reported that the tumor can be well localized. Five (05) participants estimated that 3D technology improves disease comprehension, especially with thermal sensation which could help radiologists to measure the size and the tumors’ temperature. They estimated that virtual reality combined with thermography technology could enhance diagnostic. All participants found the application easy to use. Six (06) experimenters stated that those diagnostic errors might be considerably minimized. According to some of them, the 3D representation of the tumor’s volume with thermal feedback is a novel technique for tumor localization aid. Finally, all participants said they enjoyed the encounters and learned more about the tumor. They claimed to be capable of promoting this application to their friends. The summary of the agreement level provided by medical staff is illustrated in Table 2.

## 7. Conclusions

This study’s key contribution is the development of a new approach for diagnosing lung tumors by providing thermal touch sensation using virtual reality visualization and thermal technology. This approach offered a robust platform for visually evaluating and locating tumors and measuring their temperature. The thermal properties of a lung tumor were determined from the bio-heat transfer model and were utilized to calculate tumor’s temperature evolution. The result was integrated into a proposed wearable thermo-haptic device to perceive thermal sensation when touching a lung tumor’s surface in a virtual reality environment. The proposed system was tested by a medical team who reported that thermal feedback could be useful to locate and better understand the tumor.

We should, therefore, keep working to better develop 3D visualization and localization techniques that may be applied in a variety of clinical settings. Our proposed application should also be tested on a wide number of medical professionals and patients in various context. As future work, we consider extending the validation of the proposed by collecting further lung CT images from various severity types of lesions. Moreover, we are planning to improve the proposed to merge patterns of the tumor data with real clinical scenery for the better examination, detection, and diagnosis of lung tumors. We are hoping that the advanced system will be helpful to other health scenarios facing similar challenges, including abnormalities caused by other cancers.

## Figures and Tables

**Figure 1 diagnostics-13-00567-f001:**
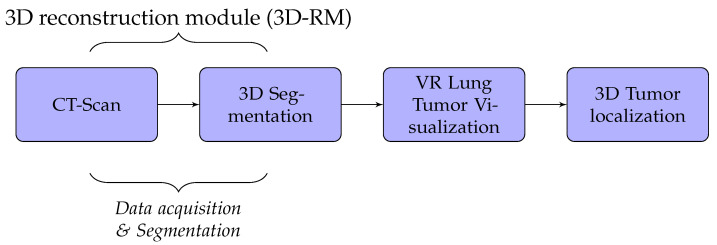
Fundamental modules mechanism.

**Figure 2 diagnostics-13-00567-f002:**
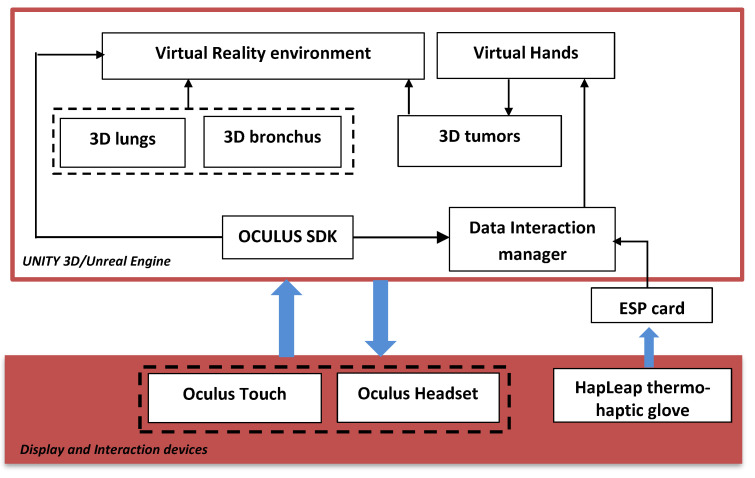
General diagram of the proposed platform.

**Figure 3 diagnostics-13-00567-f003:**
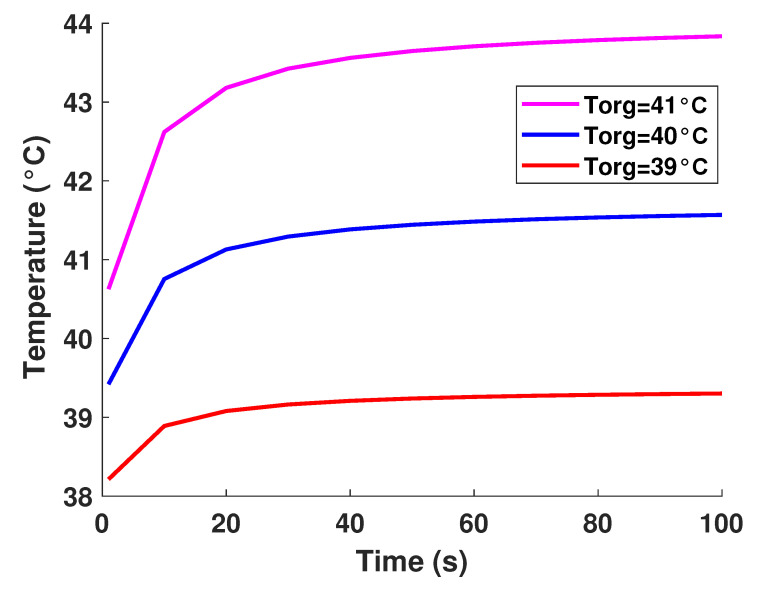
Model-based temperature profiles.

**Figure 4 diagnostics-13-00567-f004:**
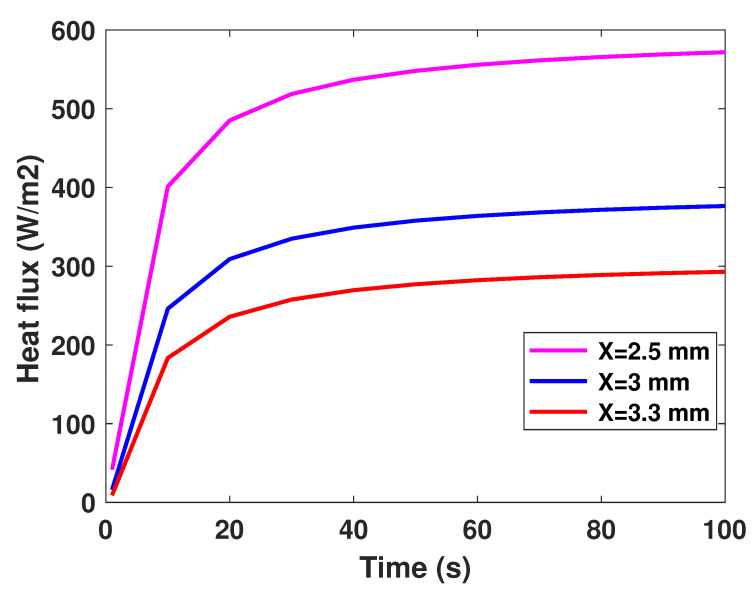
Model-based flux profiles.

**Figure 5 diagnostics-13-00567-f005:**
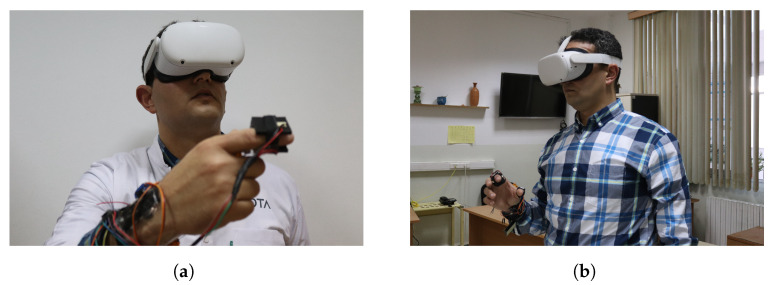
VR Integrated hardware prototype weared by a user.

**Figure 6 diagnostics-13-00567-f006:**
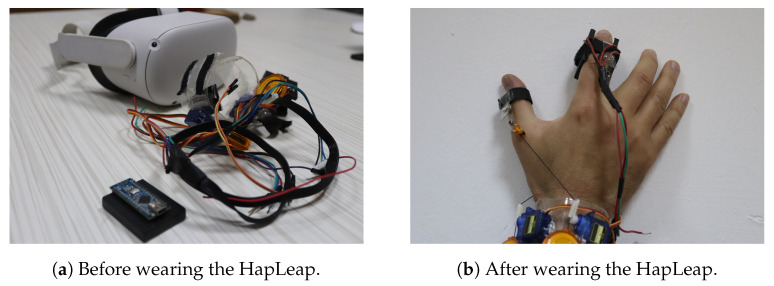
HapLeap interactive device.

**Figure 7 diagnostics-13-00567-f007:**
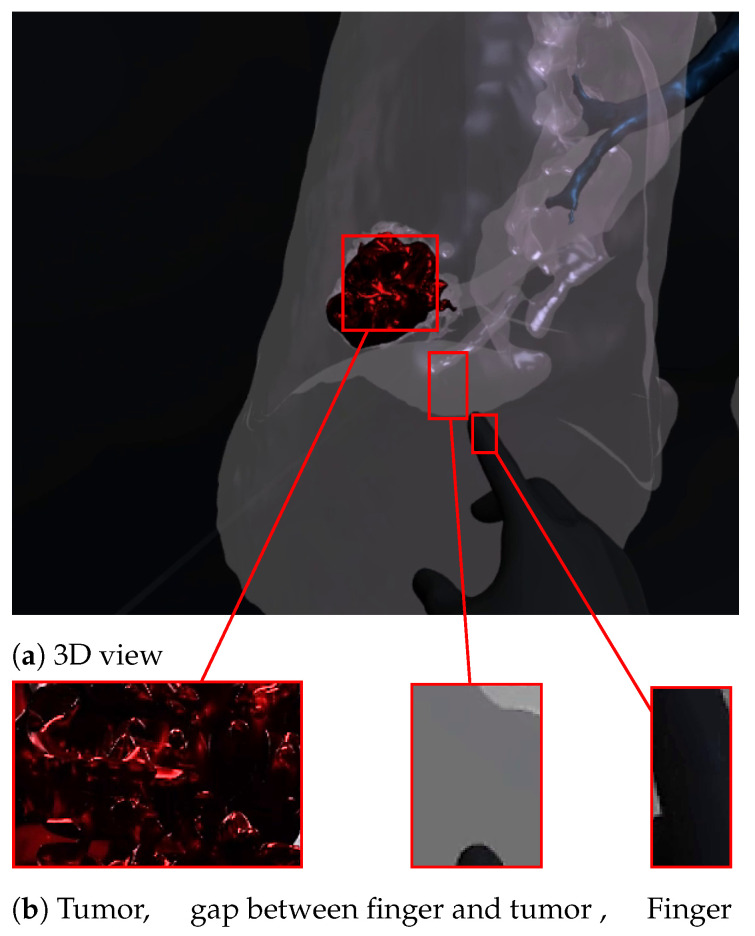
Virtual reality viewer of the tumor before touching lesion.

**Figure 8 diagnostics-13-00567-f008:**
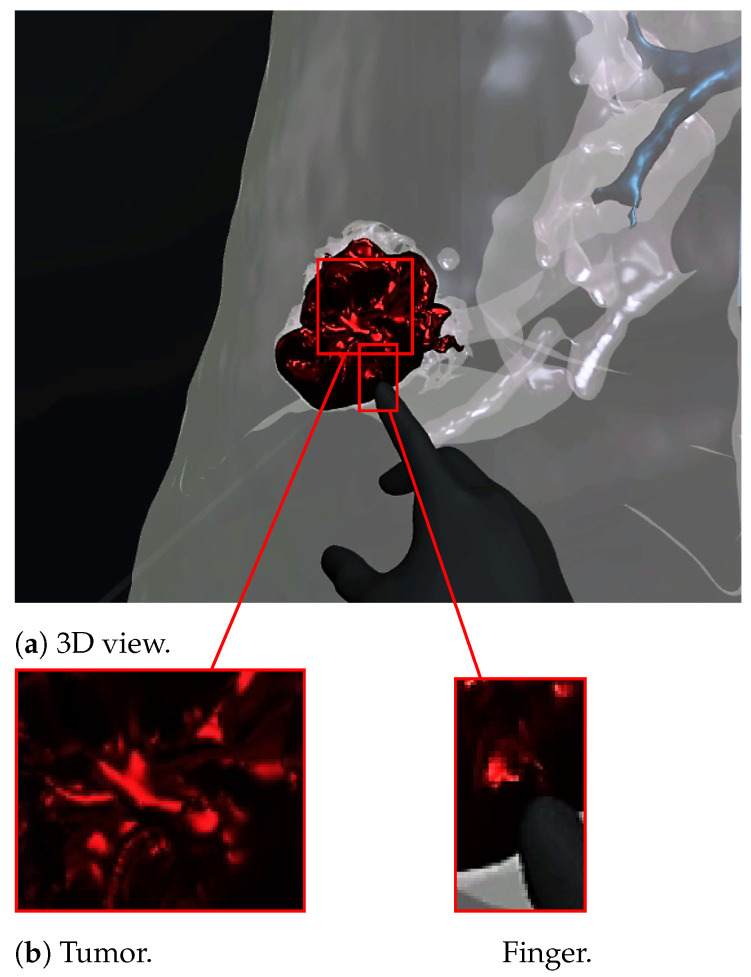
Virtual reality viewer of the finger touching and interacting with the tumor. The user feels a slight increase in his index finger’s temperature, which corresponds to the tumor’s temperature (40 °C according to [29]).

**Table 1 diagnostics-13-00567-t001:** Thermal properties of tissues and blood [28].

	**Characteristics**	
Blood	Specific heat of blood	3.64 KK/Kg/°C
	Density of blood	1000 Kg/m${}^{3}$
	Temperature of blood	37 °C
Tumor Tissue	Specific heat of frozen tumor tissue	1.23 KK/Kg/°C
	Specific heat of unfrozen tumor tissue	4.2 KK/Kg/°C
	Density of tumor tissue	1000 Kg/m${}^{3}$
	Thermal conductivity of frozen tumor tissue	$2.25*10^{-3}$ KJ/m/s/°C
	Thermal conductivity of unfrozen tumor tissue	$0.55*10^{-3}$ KJ/m/s/°C
	Density of tumor tissue	1000 Kg/m${}^{3}$

**Table 2 diagnostics-13-00567-t002:** Agreement level provided by medical staff using 7 statements.

	**Questionnaire Graded on a 7-Point Likert Scale**						
	**Strongly** **Agree**	**Agree**	**Somewhat** **Agree**	**Neutral**	**Somewhat** **Disagree**	**Disagree**	**Strongly** **Disagree**
Q${}_{1}$	0	5	2	1	0	0	0
Q${}_{2}$	4	3	1	0	0	0	0
Q${}_{3}$	3	5	0	0	0	0	0
Q${}_{4}$	1	3	1	1	1	1	0
Q${}_{5}$	2	4	2	0	0	0	0
Q${}_{6}$	2	3	1	0	0	0	2
Q${}_{7}$	5	3	0	0	0	0	0

## Data Availability

Data supporting reported results can be found at the Centre de Développement des Technologies Avancées (CDTA), Robotics and Industrial Automation Division, Algiers, Algeria.

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
