# Peer review of "Tumor Lung Visualization and Localization through Virtual Reality and Thermal Feedback Interface"

_diagnostics, 2023, doi:10.3390/diagnostics13030567_

Round 1
Reviewer 1 Report
The main proposed approach has novelty in contribution and methodology. Revision in terms of technical details is needed before publication. Also, paper organization can be improved. In this respect, some comments are suggested to describe technical details.
1. The plot of different Torg values is nearly same in the figure 3. Discuss the figure 3 with more details. What is your achievement about it?
2. Report the Figure 8 results in Table format.
3. It is suggested to discuss about runtime of each process briefly (Comparing with other methods is not needed).
4. It is suggested to review some related papers as potential applications. For example, I find a paper entitled “Detection of Lung Cancer Tumor in CT Scan Images Using Novel Combination of Super Pixel and Active Contour Algorithms”, which has enough relation. Cite this paper and some other as future works or potential applications.
5. How do you select the initial organ(tumor)temperature?
6. Review whole text in terms of possible typing errors or English mistakes. For example, Page 5, line 160, the phrase “The” should be corrected.
Reviewer 2 Report
I think the study has valuable content, and it has been comprehensive. It is a fascinating subject. To improve the scientific level of the article, the following significant corrections seem necessary.
1- This article does not have clear scientific merit or novelty. Would it be helpful if the authors clarified in the abstract how the proposed method is original and how it adds value to the article?
2- Give more details about the proposed approach, focusing specifically on its components and the relationships between them. They are the core of the solution and need more justification.
3- A significant revision is needed to the "Introduction" section to provide a more accurate and informative analysis of the pros and cons of the existing approaches and how the proposed method differs from them. A clearer explanation of motivation and contribution is also needed.
4- The conclusion sections, as well as the contribution of the study, should be supported by numerical values. To put it another way, the author should discuss future research related to the proposed method and its limitations. Why is the proposed method suitable for this unique task? In comparison with the existing approaches, what new features have the authors added to the proposed system? Clarification is needed on these points. Discussions should focus on why the proposed method is effective. It is necessary to rewrite the conclusion sections.
5- it is better to improve the logic of the introduction. You could explain the purpose and significance of utilizing VR and AR for the study of disorders in the neurobiology field. It is also possible to mention current progress and critical issues. This section can be edited by the authors using these articles. Evaluating the possibility of integrating augmented reality and Internet of Things technologies to help patients with Alzheimer’s disease - Virtual reality in the diagnostics and therapy of neurological diseases -The Role of Virtual Reality in Screening, Diagnosing, and Rehabilitating Spatial Memory Deficits- Augmented and Virtual Reality in Dermatology—Where Do We Stand and What Comes Next? - Cerebral Anatomy Detection and Surgical Planning in Patients with Anterior Skull Base Meningiomas Using a Virtual Reality Technique
Round 2
Reviewer 1 Report
The authors have given convincing answers to the questions. The method is explained in a clearer way in this version.
Author Response
We appreciate the Reviewer's insightful suggestions and comments.
Reviewer 2 Report
The manuscript was modified very well. The authors have attempted to address all of the reviewers' comments in the revised paper. The manuscript seems acceptable to me for publication in the journal with the corrections made.
Author Response
We value the Reviewer’s recommendations and comments.